# Analysis of Changes in the Opening Pressure of Marine Engine Injectors Based on Vibration Parameters Recorded at a Constant Torque Load

**DOI:** 10.3390/s23208404

**Published:** 2023-10-12

**Authors:** Marcin Kluczyk, Andrzej Grządziela, Adam Polak, Michał Pająk, Miłosz Gajda

**Affiliations:** 1Mechanical and Electrical Engineering Faculty, Polish Naval Academy, Śmidowicza 69 St., 81-127 Gdynia, Poland; a.grzadziela@amw.gdynia.pl (A.G.); a.polak@amw.gdynia.pl (A.P.); 2Faculty of Mechanical Engineering, Institute of Applied Mechanics and Mechatronics, Casimir Pulaski University of Radom, Stasickiego 54 St., 26-600 Radom, Poland; m.pajak@uthrad.pl; 3Polish Navy Headquarter, Jerzego Waszyngtona 44 St., 81-301 Gdynia, Poland

**Keywords:** vibration measurements, injectors, marine engines, diagnostics

## Abstract

This article deals with the problems related to the difficulties in the vibration diagnostics of modern marine engines. The focus was on the injection system, with a particular emphasis on injectors. An unusual approach to the implementation of research enabling the smooth regulation of the opening pressure of the mechanical injector during engine operation at a constant load was presented. This approach obtained repeatability of conditions for subsequent measurements, which is very difficult to achieve when using the classic approach that forces the injector to be disassembled after each test.

## 1. Introduction

The reliable operation of marine engines is essential for the safety of maritime transport. Very often, ship propulsion systems are equipped with only one main engine, and its inability to work may directly threaten crew members’ lives and health. Any problems related to the vessel’s manoeuvrability also affect the cargo’s safety and the marine environment.

As results from many earlier studies indicate, the components of the fuel injection systems of marine diesel engines are the most common cause of the failure of these devices. These faults, if not detected in time, can lead to engine failure and complete inoperability quickly. Considering the specific nature of the operation of marine engines, their unsuitability during a sea voyage may directly threaten the crew’s lives and health. Components of the fuel system have many parts and assemblies that are sensitive to fuel contamination [1,2]. The most sensitive elements are made with the greatest precision; injection pumps and injectors should be mentioned among them. In addition, injectors are operated in extremely unfavorable conditions, periodically exposed to very high temperatures, especially the atomizer. Diagnosing the technical condition of injectors is difficult, especially concerning engines without indicator valves (all high-speed marine engines). In such cases, to determine the correct operation of the injector, it is necessary to remove it from the engine and check its operating parameters on a unique test stand. This always means stopping the engine. The authors attempted to use vibration diagnostics to determine changes in the injector opening pressure during engine operation. For this purpose, a new approach was used to regulate the change in the opening pressure of the injector during its operation. The tests were carried out for mechanically controlled injectors. During the operation of a piston engine, a significant number of excitations interfere with the possibility of identifying the vibration signal generated by the injector [3]. In order to facilitate this task, the vibration characteristics of the injector were first tested on the injector test bench. The initially identified vibration phenomena accompanying the changing injector opening pressure were then verified during tests on a running engine.

Ships, as autonomous units, have many operation, planning, and implementation restrictions. Depending on the main propulsion used, they are stated in various defense standards, standardization standards, or manuals for individual mechanisms [4,5]. Engines of the main propulsion and generating sets on navy ships are usually medium- and high-speed engines with a capacity of over 100 kW. The measurement procedures and the assessment of the technical condition measured on non-rotating parts of piston engines of such power are described in detail in the ISO 10816:6 standard [6]. 

Based on many years of diagnostic experience gathered at the Institute of Ship Construction and Operation of the Polish Naval Academy and the analysis of the available literature, an assessment of the causes of the operational unsuitability of marine engines operated in the navy was carried out (Figure 1) [4,7,8,9,10]. The conducted analysis showed that the most frequent damage occurs in the following functional systems:Engine fuel supply system—72%;Timing system—19%;Engine air supply system—9%.

Exceeding the permissible values characterizing the operation of injection systems of marine diesel engines has two consequences from the point of view of diagnostics, i.e., it leads to an increase or decrease in the value of pressure in the engine cylinder and changes in the distribution of its values as a function of crankshaft angle (CA).

Changes in the technical condition of the fuel system not only cause a decrease in the quality of processes inside the cylinders but may also, in the long term, cause serious secondary damage to other marine propulsion system subassemblies, e.g., flexible couplings or torsional vibration dampers [11,12,13,14]. The operation of an engine in a deteriorated technical condition of an injection system always leads to a decrease in the performance and efficiency of the marine propulsion system. The reasons for this may be various [10]:Allowing low-quality fuel to be used (ineffective operation of centrifuges, contaminated filters, contaminated tanks, leaks of other substances into the fuel supply system, etc.);A lack of immediate reaction of the technical staff to minor primary damage to structural elements affecting the quality of the fuel system operation;Multiple engine starts, frequent changes in its load, and long-term operation at low loads;Chemical corrosion caused by the aggressive influence of factors contained in the fuel (sulfur and vanadium), especially during long engine downtimes;The erosive impact of solid particles contained in the fuel, as well as the cavitation erosion of flow channels;The thermal impact of a hot engine on precise dosing (e.g., sensors) and regulating elements.

The most common damage to diesel engine injectors is changes in the characteristics of the spring elements, including spring cracks. Due to the presence of solid impurities and water in the fuel, precision injector pairs also seize. As a result of the incomplete combustion of fuel, injector nozzles might be clogged. Fatigue cracks in nozzles are not very common but also occur in operational practice. Changes in the injector’s operating characteristics may also be caused by the formation of cracks in the precision pairs. More information about possible damage to injectors in diesel engines can be found in [15,16].

An essential aspect of the operation of fuel injection systems is their impact on the toxicity of exhaust gases emitted into the atmosphere [17]. As mentioned earlier, the deterioration of the technical condition of this installation always leads to the deterioration of the engine’s operating parameters, including an increase in exhaust gas toxicity. In connection with the above, intensive research on possibilities for the early detection of fuel injector failures in diesel engines, including marine engines, is conducted in many centers in Poland and worldwide [2,12,13,14,18,19,20,21]. For example, in [2], common rail fuel injectors were tested. Normally used measured points were introduced, but in postprocessing, the author used a regression coefficient, enabling the detection of problematic faults without generating a full injection map. In [13], it was confirmed that a fuel injector needle valve closing crash can excite cylinder head vibration and generate a dual-peak phenomenon. The authors in their work also indicated that this phenomenon is possible to detect only under strictly defined operating conditions of a tested engine. In [12], the authors installed sensors on the engine block and analyzed the collected signals using STFT. Conclusions were drawn that injector defects cause an increase in the value of vibration parameters in ranges above 10 kHz. Simultaneous measurements of the cylinder’s internal pressure and vibrations on the engine heads were carried out in the tests described in [21]. The authors found that, on the basis of conducted research, it is possible to determine the technical condition of injectors on the basis of just vibration signal.

The publications mentioned above-used engine structural elements as mounting locations for the transducers. In the presented publication, the authors indicate the legitimacy of introducing additional intermediary elements during the measurements, enabling a more accurate measurement of the vibration parameters of the injectors.

## 2. Materials and Methods

The tests were carried out on the laboratory stand of the Sulzer 6AL 20/24 engine. It was a 4-stroke marine diesel engine with direct fuel injection. The engine was turbocharged, and the charge air was cooled. The test object was a clockwise rotation engine with the firing sequence 1-4-2-6-3-5, equipped with a Woodward PGA multi-range speed controller.

The main problem in measuring vibration parameters concerning high-speed and medium-speed marine engines is that the injectors of these engines are located under the timing covers, and direct access to them is very difficult. Views of the injector mounting location in the cylinder head structure and the injector are shown in Figure 2.

Due to the deterioration of the mechanical properties of the springs due to the fatigue of the material from which they are made, injector opening pressure decreases, which was simulated during the tests described later in this work. The injector opening pressure change was simulated by adjusting the spring preload screw (Figure 3, Item 3). However, the change in the injector opening pressure occurs not only due to the difference in the stiffness of the retaining spring but also due to the “settling of the springs” when the seat cone and the nozzle needle wear out significantly. The problem of changing the injector opening pressure affects about 70% of cases of injector failure, and phenomena are noted after about 100 to 500 h of operation. The defects indicated above were also present in similar proportions on the injectors of the tested engine [22]. The construction and basic elements of the Sulzer 6AL20/24 engine injector are shown in Figure 3. The nozzles used in the configuration of the 6AL20/24 engines were seven-hole, closed-type nozzles with a spray angle of 159°.

The technical condition of injectors depends on many components that can be described by the following equation:(1)$$S_{wtr}=f\left(A_{r}d_{k}d_{i}\alpha_{r}h_{max}d_{s}\alpha_{i}\alpha_{st}\right)$$
where $S_{wtr}$—technical condition of the injector;*A_r_*—active cross-sectional area of spray holes;*d_k_*—diameter of the sprayer body guiding the needle;*d_i_*—leading diameter of the spire;*α_r_*—injection angle;*h_max_*—maximum stroke of the needle;*d_s_*—nozzle well diameter;*α_i_*—apex angle of the head cone;*α_st_*—cone angle of the needle seat.Examples of damage to this type of injector can be found in [23].

In all fuel systems, especially those engines with self-ignition, injectors are one of the most important elements, thanks to which we obtained a fuel–air mixture in the combustion chamber. Due to the precision of the elements made and their very difficult working conditions, especially for the nozzle, inspection, and repair works were carried out with one of the shortest service intervals.

Measurements of vibration parameters were carried out using measuring equipment and the Pulse measuring system by Bruel and Kajer. The first part used two B&K 4514B accelerometer transducers and a Bruel & Kjaer 3050-A-060 measuring cassette. Type 4514 accelerometers are titanium-piezoelectric-shear-type sensors with integral electronics. Transducers are connected with a cable by a 10-32 UNF top connector. The transducer is hermetically sealed. The type 4514 main features voltage sensitivity 10 ± 10% [mv/g], measuring range 500 [g], mounted resonance frequency 32 [kHz], amplitude response ± 10% 1–10,000 [Hz], temperature range −51 to +121 [°C], and weight—8.7 [g].

In the second part of the research, a rotational speed recorder with an optical probe of the MM024 type was additionally used. Simultaneously, with the vibration measurements on the head and injector elements, in the second part of the tests, the indication of the combustion pressure of the cylinder on which the injector opening pressure was regulated (cylinder no. 2) was also carried out. As a result of the test during which the spontaneous change of the injector opening pressure was simulated, the parameters changed, which was also visible in the indicator chart. During the measurements, only the drop in the injector opening pressure below the nominal value was simulated, i.e., a typical operating situation. According to the equations and previously conducted research, there should be a relationship [10]:(2)$$Z_{p_{otw}}\Rightarrow \left(T_{g}↑,p_{max}↓,p_{maxwtr}↓\right)\Rightarrow \left(g_{e}↑,\tau_{s}↑\right)$$
where $Z_{p_{otw}}$—decrease in injector opening pressure;$T_{g}$—gas temperature behind the cylinder;$p_{max}$—maximum combustion pressure;$p_{maxwtr}$—fuel injection pressure;$g_{e}$—specific fuel consumption;$\tau_{s}$—burning time.

It should be noted that it often happens that reducing the injector opening pressure ($Z_{p_{otw}}$) causes a slight change in the injection advance angle, along with increasing the time and dose of injected fuel. Since the drop in the injector opening pressure was not yet significant enough to significantly worsen the degree of fuel atomization, this led to masking the effect of lowering the maximum combustion pressure ($p_{max}$). 

For the first part of the tests, the injector of the tested engine’s second cylinder was removed per the manufacturer’s instructions and recommendations. After the initial inspection, the injector was mounted on the injector measurement stand to determine its vibration characteristics and the settings of the adjusting screw position corresponding to the specific values of the injector opening pressure. The injector on the test stand is shown in Figure 4a,b.

Tests were carried out on the stand for the nominal injector opening pressure of 24.5 MPa and for simulated injector failure pressures of 21.5 MPa, 18.5 MPa, 15.5 MPa, 12.5 MPa, 9.5 MPa, 6.5 MPa, 3.5 MPa, 1.5 MPa, and 1 MPa. It was established that a decrease in the pressure value of 3 MPa corresponded to a 1/4 turn of the adjusting screw. The authors’ diagnostic practice resulting from many years of measurement experience on ships shows that there are cases of fatigue cracking of nozzles or breakage of the injector spring. It also happens that due to improper installation, the nozzle comes into contact with the piston bottom. In such cases, the injector opening pressure can reach very low values. Due to the operational safety of the tested engine, pressure of 1 MPa was set only for the injector testing station. During measurements on the engine test stand, the lowest pressure we decided to set was 3.5 MPa. In the measuring station, shown in Figure 4a, to register vibration signals, two acceleration sensors were fitted as marked: the first sensor on the fuel connection pipe, the second on the screwed-in extension of the adjusting screw (Figure 4b, Item 1). By using an extension through a drilled hole in the engine timing cover, the preload of the injector spring could be adjusted during normal engine operation. Thus, the need to stop the engine was eliminated, which allowed the tests to be carried out in identical engine operating conditions with changing fuel injection conditions. In order to protect the possibility of the extension loosening during the test on the engine, a counter screw with a spring washer was used (Figure 4b, Item 2).

After completing the tests on the injector testing stand, the tested injector and all other engine injectors were adjusted and installed per the engine’s instructions and recommendations. According to the established scheme of pressure change by 3 MPa, an angle scale was determined every 90° and attached to the cylinder head. The method of fixing the scale is shown in Figure 5a. The fixing places were previously cleaned, degreased, and dried. The engine test stand general view is presented in Figure 5b. 

During the tests on the engine, vibration accelerations were measured in the following steady states of the engine (n—rotational speed; M—torque set on the hydraulic brake):n = 500 rev/min, M = 2050 Nmn = 600 rev/min, M = 3420 Nmn = 700 rev/min, M = 4390 Nm

## 3. Results

A high-pass filter with a cut-off frequency of 0.7 Hz was used during the vibration acceleration measurements. A signal sampling frequency of 65,536 Hz was used. The time courses obtained during the measurements were analyzed. First, the absolute values of the time courses of vibration accelerations were analyzed (Figure 6 and Figure 7). During the analyses, the first amplitude identifying the injector activation was omitted; then, the next five amplitudes identifying the subsequent openings of the injector were read from the pre-prepared courses; next, the average value was determined.

The waveforms in Figure 6 and Figure 7 refer to two extreme cases, i.e., nominal injection pressure of 24.5 MPa and very low pressure of 3.5 MPa. The dynamics of the injection process in these two cases were completely different. This was due to the fact that the injection pump pumped fuel with the same pressure in both cases and the behavior of the actuator element, which, in this case, was the injector, was different. This included the amplitudes of recorded vibration accelerations and the duration of the injection process itself. It was much longer in the case of low injector opening pressure due to the significantly greater lift of the needle resulting from the lower pre-tension force of the injector needle.

The data collected in this way are presented graphically in Figure 8. The figure presents the values of accelerations for the whole series of measurements from 24.5 to 1 MPa. The change trend as a function of the injector opening pressure change shows a decrease in the vibration acceleration values recorded by both sensors. Therefore, the measurements provided diagnostically useful information in this regard.

Spectra of vibration accelerations were also made for each series of measurements (Figure 9). Broadband analysis using the calculation of the average RMS value of vibration accelerations indicated that it was possible to find the range of frequencies sensitive to changes in the injector opening pressure. In the considered case, this range was 300–2300 Hz for the measuring point located on the injector adjustment screw. Detailed results of average acceleration values for this frequency range are presented on the right side of Figure 9 (delta cursor values). 

The first part of the research noticed dependencies that could be used as a diagnostic method. It was possible to detect pressure changes by analyzing the vibration parameters. For the time courses, this was indicated by the dependence on decreasing vibration acceleration values as a function of increasing injector deregulation.

### Test Results Recorded during Tests on a Marine Engine Stand

The next part of the research was the measurements of the engine. The change in the injector opening pressure was regulated from 24.5 MPa to 3.5 MPa for each rotational speed setting (500, 600, 700 rev/min). Analyzing graphs from time courses and frequency spectra also requires knowledge of the engine design, e.g., the timing system. Also helpful in the analysis of time courses is the imposition of a synchronized signal from the tachometer on the graph, which indicates the upper position of the piston on one of the cylinders. An exemplary chart is shown in Figure 10. Thanks to such a procedure, it was possible to precisely determine the successive extortions associated with the operation of the tested injector. Information about the position of the piston would make this task significantly more difficult.

Measurements in the second part of the study were made using three accelerometers. In addition to the points selected in the first part of the research, a point located on the tie bolt of the head was also selected. An example time course for the injector opening pressure of 24.5 MPa, n = 500 rev/min, M = 2050 Nm is shown in Figure 11.

Using the previously described methodology, the average vibration acceleration values were determined for all injector opening pressures recorded during engine operation (Figure 12).

For each of the rotational speeds, collective indicator charts were also made, confirming the dependence of the impact of the injector opening pressure on pressure changes in the combustion chamber. Figure 13 shows a graph of changes in cylinder pressure for a constant rotational speed n = 500 rev/min and torque set on the hydraulic brake M = 2050 Nm with the simultaneous regulated opening pressure of the injector.

With regard to the time courses, the same way of presenting the results for the tests carried out at other rotational speeds was adopted. The results for 600 rev/min are given below (Figure 14). Figure 15 shows the waveforms of the indicated pressure accompanying the change of the injector opening pressure. Figure 16 and Figure 17 correspond to a rotational speed of 700 rev/min.

The analysis of Figure 12, Figure 14 and Figure 16 indicated an increase in the intensity of the recorded vibration parameters with increasing rotational speed and torque load on the engine. This was a consequence of the greater dynamics of the operation of the injection system, which, at higher rotational speeds, has to perform its task in a shorter time. Interestingly, these changes ran differently depending on the measurement point of vibration accelerations. And so, for the speed of 500 rev/min, the most appropriate point seemed to be the high-pressure pipe supplying the injector—this is probably due to the large amplitudes related to fuel pulsation in the pipe, and not the injector operation itself. On the other hand, for the speeds of 600 and 700 rev/min, the best results were obtained for measurements on the extension of the injector adjustment screw. It should be emphasized here that the results gathered in relation to the measurements on the head tightening bolt where the tested injector was installed did not show useful features for diagnosing the injector’s technical condition. It should be stated that preparations for vibration diagnostic tests of marine piston engine injection systems must always be preceded by a thorough analysis of the structure and preliminary vibration tests. This is to determine the most appropriate engine operating ranges during tests and select potential measurement points to obtain the most sensitive diagnostic parameters.

## 4. Discussion

High and medium-speed marine engines currently have a very compact design, and the injection system components are difficult to access. Unfortunately, they are often not equipped with indicator valves, making measuring the pressure inside the cylinder impossible. These limitations force diagnosticians to look for new methods of measuring and assessing the technical condition of components. The tests confirm the possibility of making small modifications for measurements to facilitate their reliable implementation. The diagnostic method using only the indicated pressure to assess the state of the parameters of a single cylinder is not sensitive to changes in the injector opening pressure. Even when the opening pressure drops to 3.5 MPa, the maximum indicated pressure remains close to the values obtained for a technically efficient injector. The higher the engine speed, the more pronounced this situation (see Figure 14, Figure 16 and Figure 17). This results from the fact that at higher rotational speeds, the temperature increase per unit of time increases. Thus, the quality of fuel atomization by a malfunctioning injector starts to play a secondary role at the same time as the speed increases.

The conducted research and analyses show that the most effective method of detecting a simulated malfunction is the analysis of time waveforms. The measurements made on the injector test stand indicate an almost linear dependence of the recorded accelerations on the change in injector opening pressure. The results of tests on an engine test stand during its operation in steady-state conditions (Figure 12, Figure 14 and Figure 16) confirm the results of the tests performed in the first stage of research conducted on an injector test stand (Figure 8). The results present sensitivity to a change in injector opening pressure greater than 6 MPa. These changes are observable in particular for rotational speeds closer to the rated motor speeds. In terms of reducing the injector opening pressure to values lower than 6 MPa below the nominal value, the vibration parameters do not show a clear relationship with the technical condition of the injector. It should be emphasized here that changes in the injector opening pressure also slightly affected the course of the maximum indicated pressure. The FFT method’s analysis of the amplitude spectra confirmed the presence of frequencies sensitive to changes in the injector opening pressure only in the signals recorded in the first part of the research. A possible reason for not isolating such frequencies from the signals from a series of measurements on the engine is the excessive impact of vibration excitations related to the operation of the engine’s crank and piston system—all main and auxiliary processes occur for each cylinder with the same frequency.

The applied approach of changing the mechanical pressure of the fuel injector during engine operation allowed for the collection of unique data. Their analysis confirms the possibility of implementing vibration methods in diagnosing malfunctions of marine piston engine injectors, with particular emphasis on malfunctions related to changes in injector opening pressure.

## Figures and Tables

**Figure 1 sensors-23-08404-f001:**
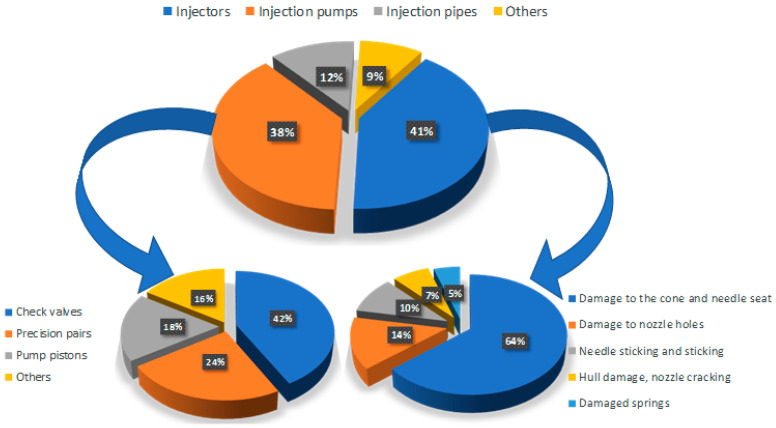
Percentage share of failures of individual elements of the fuel system [10].

**Figure 2 sensors-23-08404-f002:**
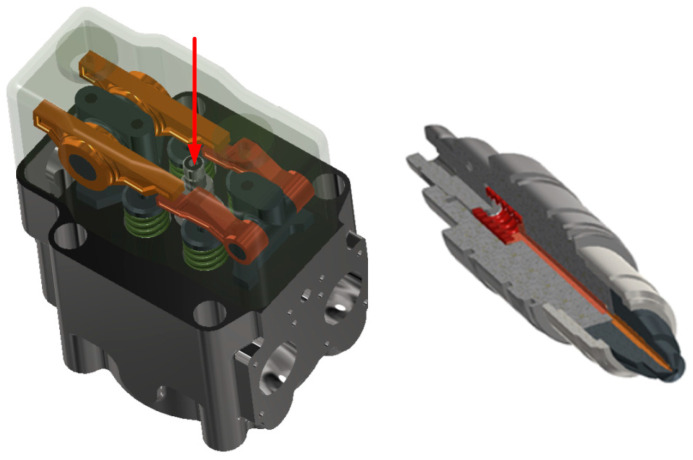
View the tested engine’s complete cylinder head (**left**). The red arrow marks the injector mounting location. A quarter cross-section of the tested injectors (**right**).

**Figure 3 sensors-23-08404-f003:**
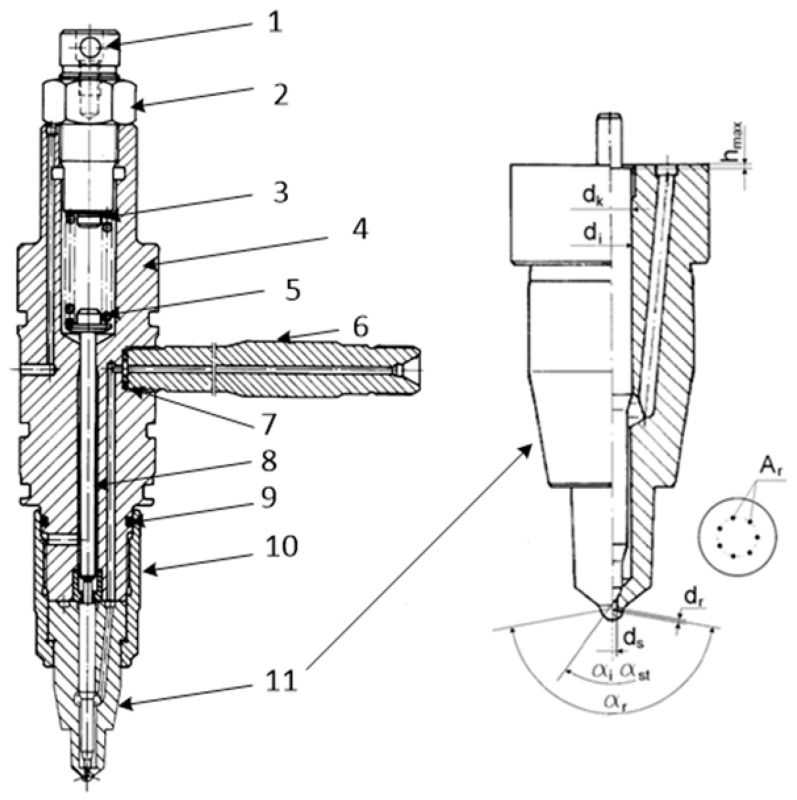
Engine injector 6 AL 20/24. 1—adjusting screw, 2—nut, 3—spring, 4—injector holder, 5—spring plate, 6—fuel connector, 7—gasket, 8—pin, 9—sealing ring, 10—union nut, 11—nozzle [12].

**Figure 4 sensors-23-08404-f004:**
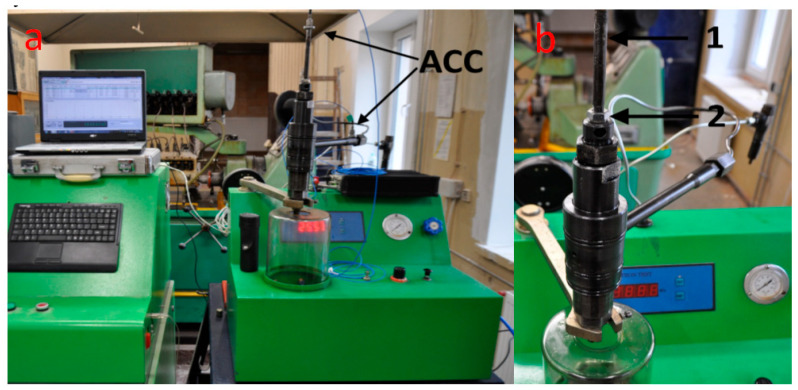
(**a**) Injector during testing on the injector test stand; (**b**) details of the solution allowing the installation of the accelerometer while the engine was running. ACC—accelerometer, 1—screwed-in extension of the injector adjusting screw 2—counter screw with a spring washer.

**Figure 5 sensors-23-08404-f005:**
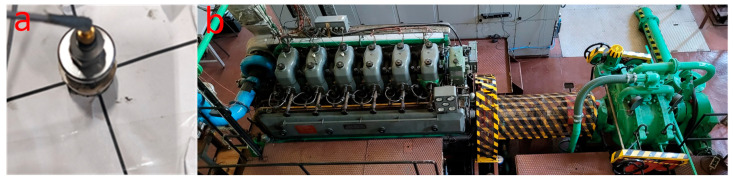
(**a**) View of the accelerometer mounted via an extension on a working engine; (**b**) view of the Sulzer 6AL 20/24 engine test stand.

**Figure 6 sensors-23-08404-f006:**
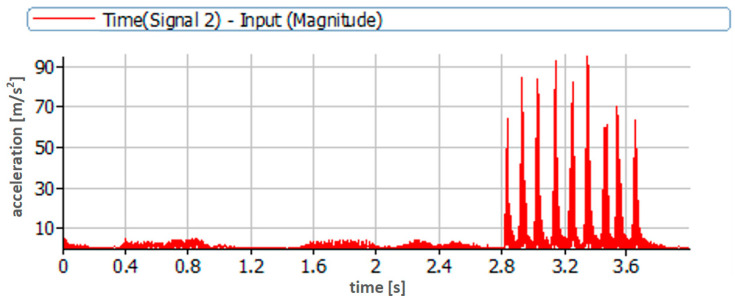
Time course of vibration acceleration recorded during injector operation with nominal opening pressure of 24.5 MPa—measurement point located on the adjusting screw.

**Figure 7 sensors-23-08404-f007:**
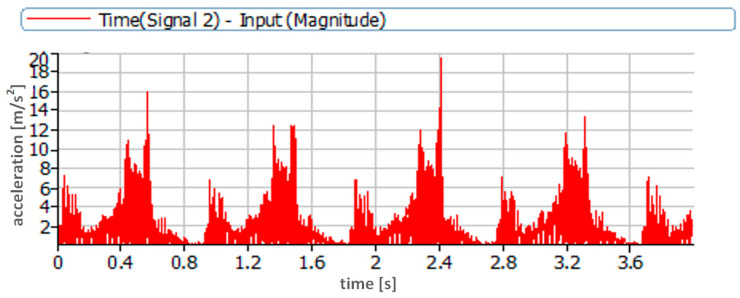
Time course of vibration acceleration recorded during injector operation with nominal opening pressure of 3.5 MPa—measurement point located on the adjustment screw.

**Figure 8 sensors-23-08404-f008:**
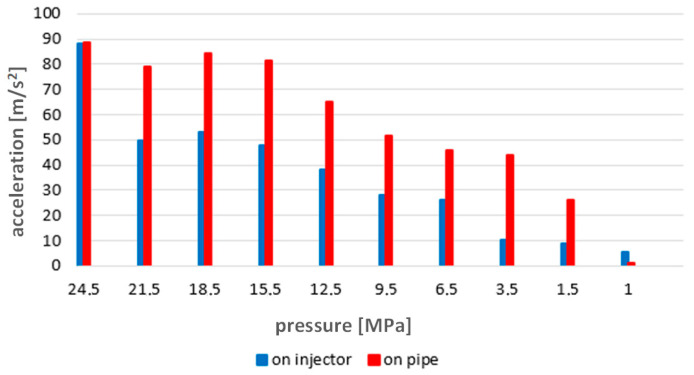
Graph of average vibration acceleration values for all injector opening pressures recorded on the injector test stand.

**Figure 9 sensors-23-08404-f009:**
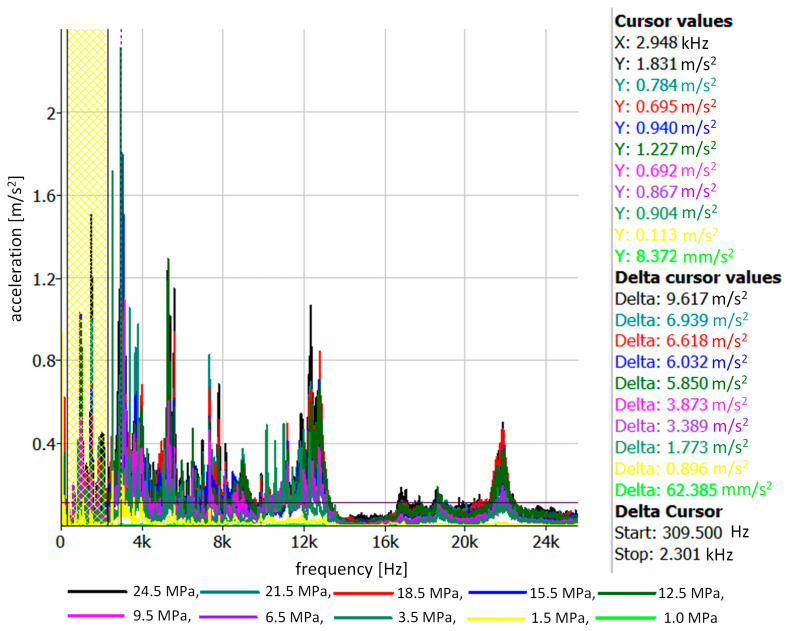
Spectrum of vibration accelerations for all injector opening pressures recorded on the injector test stand, with the marked value of the average vibration acceleration in the range of 310–2300 Hz.

**Figure 10 sensors-23-08404-f010:**
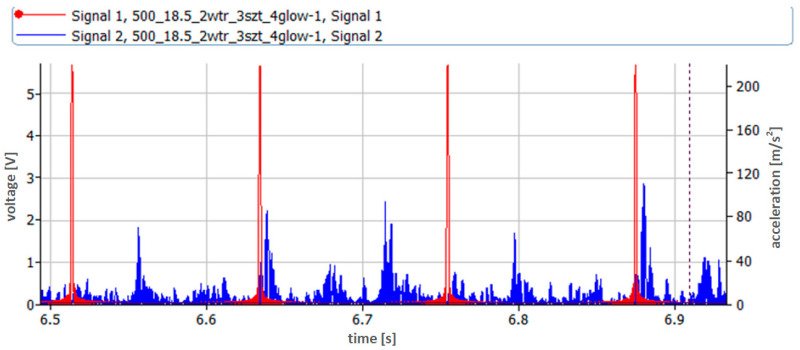
Time course with superimposed tachometer signal with an indication TDC of first cylinder.

**Figure 11 sensors-23-08404-f011:**
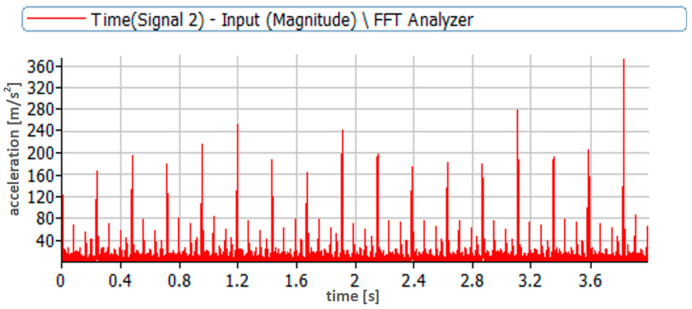
Time course for injector opening pressure 24.5 MPa, engine test, n = 500 rev/min, M = 2050 Nm.

**Figure 12 sensors-23-08404-f012:**
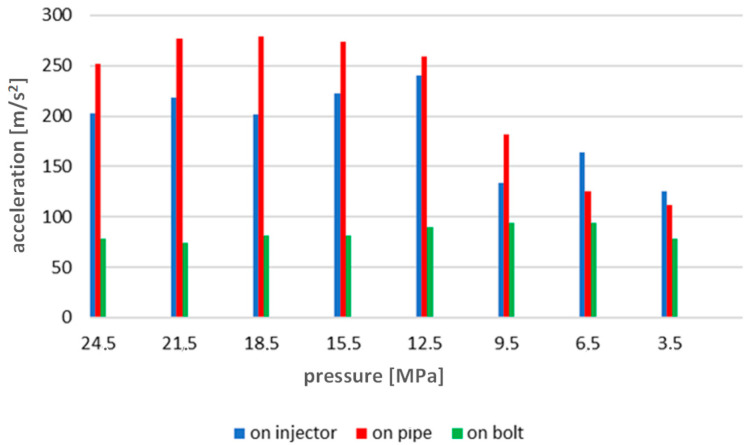
Graph of average vibration acceleration values for all injector opening pressures recorded during engine operation, engine shaft speed 500 rev/min.

**Figure 13 sensors-23-08404-f013:**
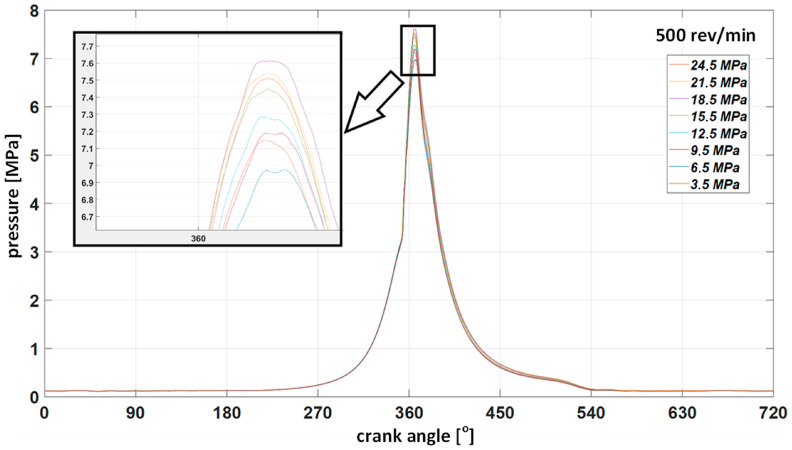
Collective indicator graph for injector opening pressure from 24.5 to 3.5 MPa at 500 rev/min.

**Figure 14 sensors-23-08404-f014:**
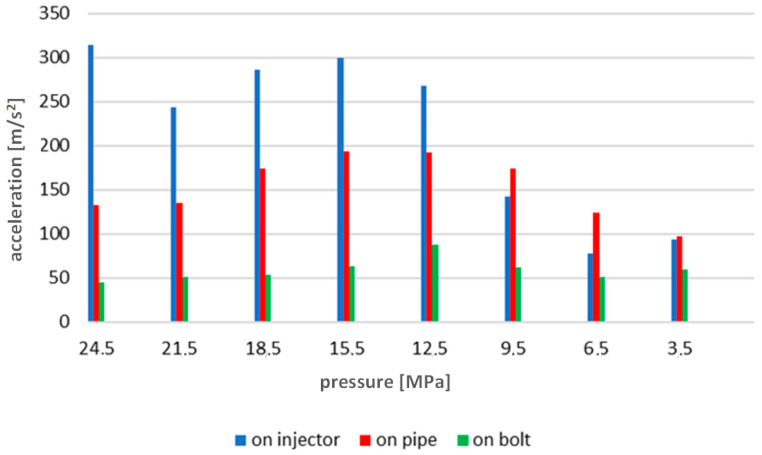
Graph of average vibration acceleration values for all injector opening pressures recorded during engine operation, engine shaft speed 600 rev/min.

**Figure 15 sensors-23-08404-f015:**
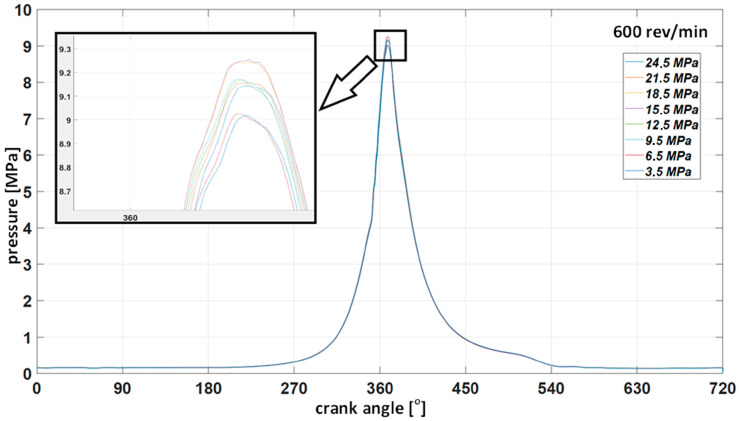
Collective indicator graph for injector opening pressure from 24.5 to 3.5 MPa at 600 rev/min.

**Figure 16 sensors-23-08404-f016:**
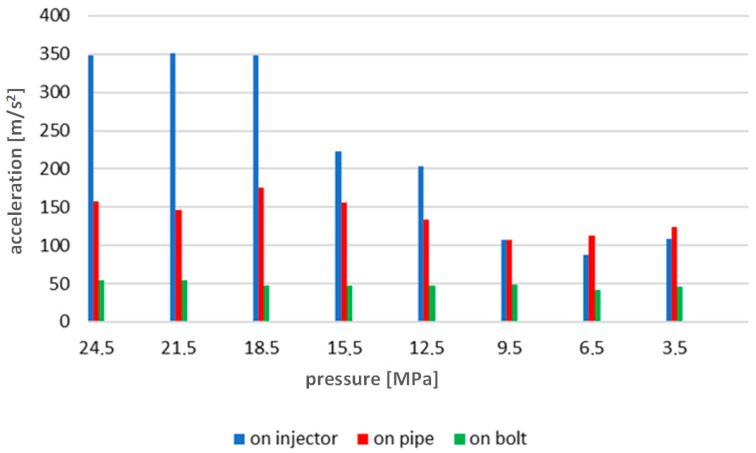
Graph of average vibration acceleration values for all injector opening pressures recorded during engine operation, engine shaft speed 700 rev/min.

**Figure 17 sensors-23-08404-f017:**
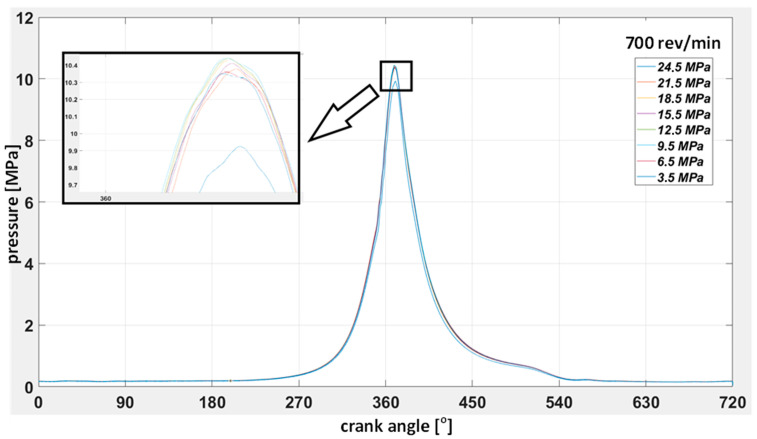
Collective indicator diagram for injector opening pressure from 24.5 to 3.5 MPa at 700 rev/min.

## Data Availability

Data available upon request from the corresponding author.

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
