# Peer review of "Analysis of Changes in the Opening Pressure of Marine Engine Injectors Based on Vibration Parameters Recorded at a Constant Torque Load"

_sensors, 2023, doi:10.3390/s23208404_

Round 1

Reviewer 1 Report

1. Firstly, the similarity of the manuscript is 26%. It has to be reduced. 

2. Secondly, the novelty of the manuscript is not understood. Especially, in the last paragraph of the Introduction, there are repeated sentence matching. Hence, I didn't check.

3. The title of the manuscript is not clearly written.  

4. The analysis of the results is to be performed more in a quantitative way. 

A technical manuscript should be written in the third person. 

Author Response

Dear Reviewer, at the beginning we would like to thank you very much for your valuable comments, they are very accurate and we fully agree with them. We would like to apologize that response took us so long. Detailed responses to individual comments are provided in the attached file

Reviewer 2 Report

Dear Authors,

Due to the subject matter the article is up-to-date and interesting. However, taking into account the errors it contains, in my opinion the manuscript in this version should not be accepted for publication. In order to improve the quality of the article, I propose to take into account detailed issues:

1)      Abstract: It should emphasize the novelty of the manuscript in relation to previously published works.

2)      Introduction: A closer look at the methods for diagnosing diesel engine injection systems presented in other works would be beneficial.

3)      Material and Methods: Line 128 – The authors refer to figure 3 rather than figure 4?

4)      Material and Methods: Equation (1): Swtr explanation is missing?

5)      Material and Methods: Line 168 – Zpotw should rather be referred to as "decrease of injector opening pressure". This is because the change can be up or down, which entails a different effect on engine performance.

6)      Material and Methods: Figure 6 - The photo does not show the entire image. It would be more beneficial to replace it with one that shows a view of the engine under test or add a second photo next to it showing that view.

7)      Results: Why was such a range of injection pressure drop adopted? Aren't low values, such as 1MPa, too small considering the operation of the engine (possibility of adequate fuel atomization and self-ignition?).

8)      Results: The vertical axis captions are missing in Figures 7 through 18 and the horizontal axis captions (Figures 9,10,11,13,15 and 17). In addition, units are missing when describing the horizontal axis (Figures 14, 16 18).

9)      Results: Figure 7 - why is there such a long delay in recording maximum acceleration amplitudes?

10)   Results: Figure 9 – how were the average values obtained? - comparing them with the values in Figures 7 and 8 seem too high?

11)   Results: Figures 14, 16, 18 - are averaged pressure waveforms shown on the indicator charts (if so, for how many duty cycles)?

12)   Results: In the description in the article (similarly in Equation 2), it is noted that one of the parameters that decreases with a decrease in injection pressure is pmax - maximum combustion pressure. Meanwhile, in the indicator charts (figures 14, 16 and 18), this is not consistent, for example, in figure 14, the maximum pressure was obtained for an injection pressure of 18.5 MPa. Similarly, in Figure 16, pmax was obtained for 15.5 MPa. Why?

13)   Results: Figures 14, 16, 18 - It would be beneficial for the reader to use the same line colors for each injection pressure on each graph.

14)   Results: Lines 305-317 - Analyzing the results, the authors emphasize that the appropriate ones are those characterized by the highest values of vibration acceleration (for engine speed of 500 rpm - "the high-pressure pipe supplying the injector"; for speeds of 600 and 700 rpm "the best results were obtained for measurements on the extension of the injector adjustment screw." However, in a diagnostic context, the changes in amplitudes should be evaluated taking into account the features of the diagnostic parameter and not the maximum values - so this still needs to be detailed and clarified.

15)   Discussion: Lines 336-342 – “Tests in steady-state conditions with the engine running confirm the results of the tests performed in the first stage and show sensitivity to  a  change  in  injector  opening  pressure  greater  than  6  MPa.  These  changes are observable in particular for rotational speeds closer to the rated motor speeds.  In terms of reducing the injector opening pressure to values lower by 6 MPa than the  nominal value, the vibration parameters do not show a clear relationship with the technical condition of the injector.”  - The conclusion is not understood by the reader. For which vibration parameters during engine tests, can be confirmed to be consistent with the bench tests of the injector, the so-called "first stage".

16)  The Conclusions of the entire article is missing.

Best regards

The manuscript contains editorial and language errors (e.g., Material and Methods: Line 128 - a space is missing between “to” and “¼ turn”; Results: Line 216 - a space is missing between "." and "First"; commas for injection pressure values in the legend descriptions of Figures 14, 16 and 18 should be replaced by periods; Item 13 in the literature list refers to two sources). I suggest a proofreading of the manuscript.

Author Response

(The authors gave the same response as above.)

Reviewer 3 Report

This paper focus on the vibration diagnostics of modern marine engines with a special emphasis on the injectors, and in particular on the smooth regulation of the opening pressure of the mechanical injector when the engine operates at constant load. The authors are suggested to focus on the following points before a final decision can be made:

1.- Introduction section. The authors are suggested to detail and explain the type of faults in the injection system and the ways to identify such faults.

2.- Figure 4. Are these failures the most common? Are there other failure modes not presented in the photographs? Do you have permission to reproduce these photographs from reference [22]?

3.- The authors are suggested to provide all the characteristics of the accelerometers used  (sensitivity, range, …).

4.- The authors are suggested to improve Figure 5. Does ACC mean accelerometer? The authors are also suggested to add a photograph with only the accelerometer.

5.- The accelerometers shown in Fig. 5b seem bulky enough to affect the vibration frequency and modes of the injector. The authors are suggested to develop this point.

6.- Results section. The graphs presented must be discussed. For example why at lower pressure the pattern of the vibration is more stable and the amplitude is lower that at higher pressure.

7.- The statement in lines 228-229 is not true for the whole analyzed pressure range, as shown in Figure 9. Please develop and explain why the decrease is not monotonic.

8.- Figure 10 is chaotic. The authors are suggested to add a Table to extract useful information.

9.- The authors have obtained interesting information about the injection system. However, it is not clear how to go further and to apply this system in a real world application. This is a critical point of this work that must be addressed in a specific section.

10.- references must be updated and the format corrected. There is a clear lack of JCR journal references from 2022-2023.

I hope that the remarks above will help to improve quality of the paper.

I suggest proofreading the paper carefully.

Author Response

(The authors gave the same response as above.)

Round 2

Reviewer 1 Report

All corrections are performed appropriately. 

Reviewer 3 Report

The authors have replied all my questions

The authors are suggested to proofread the paper